# Mapping Restoration Activities on Dirk Hartog Island Using Remotely Piloted Aircraft Imagery

**Lucy Wilson [1,\*], Richard van Dongen [2], Saul Cowen [2,3] and Todd P. Robinson [1]**

[1] School of Earth and Planetary Sciences, Curtin University, Perth, WA 6845, Australia; t.robinson@curtin.edu.au

[2] Department of Biodiversity, Conservation and Attractions, 17 Dick Perry Avenue, Kensington, WA 6151, Australia; ricky.vandongen@dbca.wa.gov.au (R.v.D.); saul.cowen@dbca.wa.gov.au (S.C.)

[3] School of Biological Sciences, University of Western Australia, 35 Stirling Highway, Crawley, WA 6009, Australia

[\*] Correspondence: lucy.wilson1@postgrad.curtin.edu.au

**Abstract:** Conservation practitioners require cost-effective and repeatable remotely sensed data for assistive monitoring. This paper tests the ability of standard remotely piloted aircraft (DJI Phantom 4 Pro) imagery to discriminate between plant species in a rangeland environment. Flights were performed over two 0.3–0.4 ha exclusion plot sites, established as controls to protect vegetation from translocated animal disturbance on Dirk Hartog Island, Western Australia. Comparisons of discriminatory variables, classification potential, and optimal flight height were made between plot sites with different plant species diversity. We found reflectance bands and height variables to have high differentiation potential, whilst measures of texture were less useful for multisegmented plant canopies. Discrimination between species varied with omission errors ranging from 13 to 93%. Purposely resampling *c.* 5 mm imagery as captured at 20–25 m above terrain identified that a flight height of 120 m would improve capture efficiency in future surveys without hindering accuracy. Overall accuracy at a site with low species diversity ($n = 4$) was 70%, which is an encouraging result given the imagery is limited to visible spectral bands. With higher species diversity ($n = 10$), the accuracy reduced to 53%, although it is expected to improve with additional bands or grouping like species. Findings suggest that in rangeland environments with low species diversity, monitoring using a standard RPA is viable.

**Keywords:** species differentiation; remote sensing; UAV; monitoring and evaluation; rangelands

## 1. Introduction

### 1.1. Remotely Piloted Aircraft Environmental Monitoring

Australian conservation monitoring is often conducted manually at the plot-scale [1]. Satellite remote sensing as an assistive dataset for testing plot-scale conservation efficacy has been largely untapped due to previous impediments, such as prohibitive costs or acquisition of capture with insufficient resolution. Remotely Piloted Aircrafts (RPAs) have begun to fill the void between open access satellite imagery with moderate resolution and very high resolution satellite imagery, too costly for repeat monitoring of vegetation changes at the plot-scale [2]. There is an urgent need for guidelines to assist conservation practitioners with best-practice RPA capture and subsequent processing [3].

Modern RPAs can capture imagery with subcentimetre spatial resolution [2,4] and have been used to monitor individual plants and grasses [5]. Differentiation of a single target species by timing imagery acquisitions with some distinguishing feature (e.g., flowers, leaf colour, defoliation) relative to coexisting species is important for tracking the trajectories of individual species (e.g., [6]), though it is less useful for studies that require quantification of multiple species with different phenological cycles. Studies that attempt to differentiate numerous species from very high resolution (VHR) remotely sensed imagery

vary in heterogenous environments and could benefit from having a universally applicable approach (e.g., [3,7]).

### 1.2. Object-Based Image Analysis

Traditional methods of classifying VHR aerial imagery focus on the spectral reflectance per pixel [8–10]. However, it has been shown that pixel-based approaches to classification may produce suboptimal results with RPA imagery [11–14]. Instead, RPA imagery benefited by being complemented with Object-Based Image Analysis (OBIA) techniques [15]. OBIA groups homogenous raster pixels into objects or segments [16], which are then classified as a unit [17]. In general, OBIA is considered to be an improvement over traditional per-pixel techniques due to pixel adjacency considerations [18], a reduction in positional inaccuracies [19], mitigation of the 'salt-and-pepper' phenomena [20], and for identifying features with high spectral variability within classes [14].

Commonly, the segment is bounded to the object being delineated, e.g., individual land cover areas or entire tree canopies [10,18]. However, it is not feasible to bind the segment to the whole canopy where rangeland plant species coexist in space and low-lying growth forms intertwine. Here, we propose that multisegmentation of the individuals within an image may overcome this challenge.

### 1.3. Dirk Hartog Island History

Dirk Hartog Island, on the west coast of Western Australia, was used by pastoralists to farm sheep between *c.* 1869 to 2009, which, combined with the presence of exotic ungulates (goats) and feral cats, grossly altered the landscape ecology, including the local extinction of at least eleven vertebrate taxa [21–23]. In 2009, the island was established as a national park by the Department of Biodiversity, Conservation and Attractions [24]. To restore biodiversity and ecosystem services, the Dirk Hartog Island National Park Ecological Restoration Project ('Return to 1616') was initiated in 2011, with a vision to return the island to its original state when Dutch explorers first visited in 1616 [25]. Sheep, goats, and feral cats were removed or eradicated by 2018 [23,26]. This subsequently allowed for the successful conservation introduction of two threatened macropod species, the rufous and banded hare-wallaby (*Lagorchestes hirsutus* and *Lagostrophus fasciatus*, respectively) [27]. Additional translocations are planned or underway with the aim of restoring the island's former fauna assemblage, which may result in beneficial conservation outcomes for some threatened species, e.g., the dibbler (*Parantechinus apicalis*). Other species, such as the boodie (*Bettongia lesueur*) and Shark Bay bandicoot (*Perameles bougainville*), may also help restore ecosystem services through their digging activities [28,29]. Identifying the flow-on effects that these changes may have on the island's biodiversity and ecosystem function requires a framework for repeat environmental monitoring. These methods should allow for both climax and pioneer plant species to be mapped.

### 1.4. Objectives

An operational remote sensing approach would greatly assist monitoring flora species composition and change on Dirk Hartog Island as it recovers. This study examines the potential of RPA imagery for monitoring plant species at the plot-scale. We have the following aims: (a) identify a suite of image-derived variables suitable for species-level discrimination and explore if these change between study sites, (b) provide a method for determining optimal height for practitioners, and (c) compare the classification measurements of two study sites, differing in species richness, using canopy multisegmentation.

## 2. Materials and Methods

### 2.1. Study Area

Dirk Hartog Island is situated on the Gascoyne coast of Western Australia (central point: 25°50′ S 113°05′ E; Figure 1A). It is approximately 630 km$^2$, 80 km long, and up to 12.5 km wide (Figure 1B). Denudation from prolonged erosion and the absence of tectonic

activity has resulted in a low-lying landscape, approximately 180 m above sea level at the highest point [30]. The soil is predominantly composed of wind-transported aeolian sand and contains carbonate grains from biogenic materials [31]. It experiences an annual semi-arid climate [32,33]. Average annual rainfall from the weather station at Denham, *c.* 40 km away, was estimated to be 204.6 mm per year between 2000 and 2019 [34]. The minimum and maximum annual temperature is approximately 18 °C and 27 °C, respectively. The island supports five vegetation communities: hummock grassland, low closed heath, low open heath, low open shrublands, and tall open heath [35].

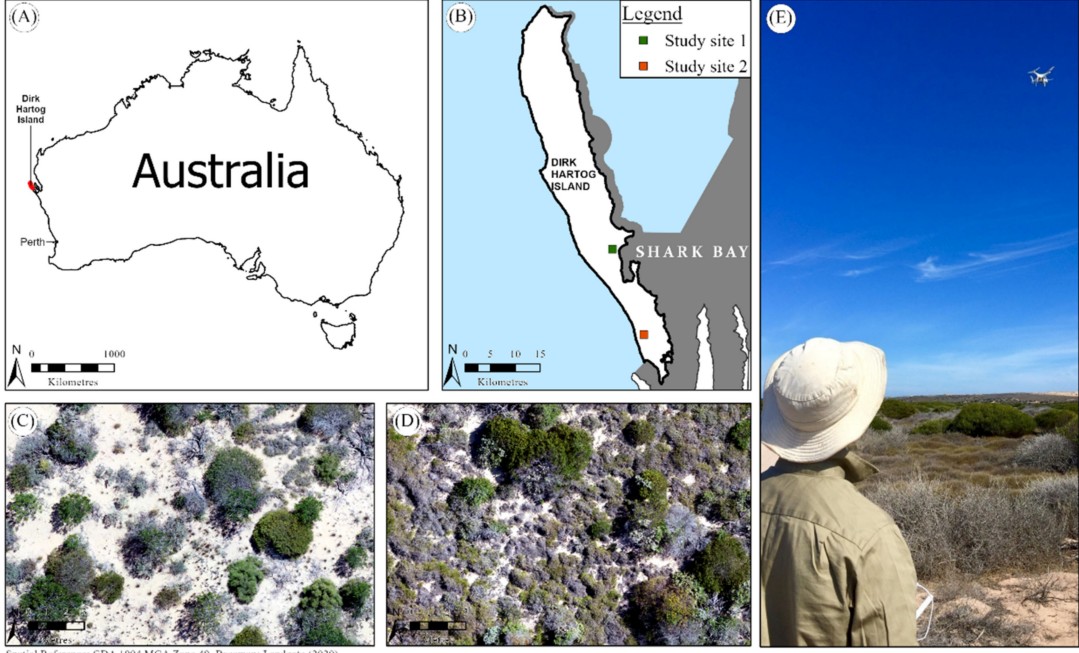

**Figure 1.** Map showing the location of (**A**) Dirk Hartog Island and (**B**) study sites in relation to Dirk Hartog Island. An exemplar of the aerial imagery and species compositions found at (**C**) study site 1 and (**D**) study site 2. (**E**) Researcher flying remotely piloted aircraft to capture Dirk Hartog Island (Photo: Lucy Wilson).

Exclusion plots (40 m × 40 m; fencing 90 cm high) were erected for the purpose of protecting vegetation from ground-dwelling fauna for future studies measuring change. Given that vegetation is still in recovery from grazing/browsing by ungulates, exclusion plots retain this process without being confounded by the activity of translocated fauna. The two study sites used for analysis comprise an exclusion plot location and peripheral areas, hereby referred to as study sites 1 and 2 (Figure 1B). The northernmost site (study site 1) is *c.* 0.4 ha (Figure 1C), whilst study site 2 is *c.* 0.3 ha (Figure 1D).

### 2.2. Reference Data and Study Species

Aerial imagery (RGB) was captured using a DJI Phantom 4 Pro RPA on 16 September, 2018 with a radiometric resolution of 8 bits. A flight plan was prerecorded in Litchi version 4.14.0 g for drone navigation in the field [36]. The starting flight height above ground for study site 1 was 22.1 m, resulting in a pixel resolution of 5.52 mm. For study site 2, the flight height was 25 m with a pixel resolution of 6.06 mm. RPA imagery tiles were mosaicked into a single image for each study site using Photoscan-pro version 1.4.2 [37].

Field surveys were conducted in autumn between 29 April 2019 and 1 May 2019. A Samsung Android tablet was used to display the georectified RPA images as an assistive tool for mobile field collection. Plants that could be unambiguously matched in the field to the image were delineated, and each individual was photographed and given an incremental photo number. A unique identifier (ID) was recorded for each sample using a composite of

the site ID and the photo number. The centroid coordinates of each sample were recorded (Zone 49, Map Grid of Australia 1994) along with the species name and general notes. Vegetation was identified to the species level by the field team and later confirmed by double-blind assessment from photo assessments by two botanists familiar with plant species of the area.

Plant samples ranged from pioneer to climax individuals, and their respective canopy boundaries were digitised using ArcGIS Pro version 2.4.0 [38]. Canopy extent, without exceeding the boundary, was approximated from the RPA imagery. Plant canopy boundaries and segmented datasets were overlaid in RStudio 1.2.1335 executing R version 3.6.1 [39,40] for later training of the machine learning algorithm.

A total of 174 individual plants across 9 species were sampled at study site 1 (Figure 2). Study site 2 sampling totalled 82 individuals across 4 species. This included three species that were found at both sites: *A. ligulata*, *A. vesicaria*, and *T. plurinervata*. Vegetation comprised greater vegetated ground coverage and less exposed earth in study site 2 than 1. Numbers of plants found in each study site can be found in Figure 2 caption.

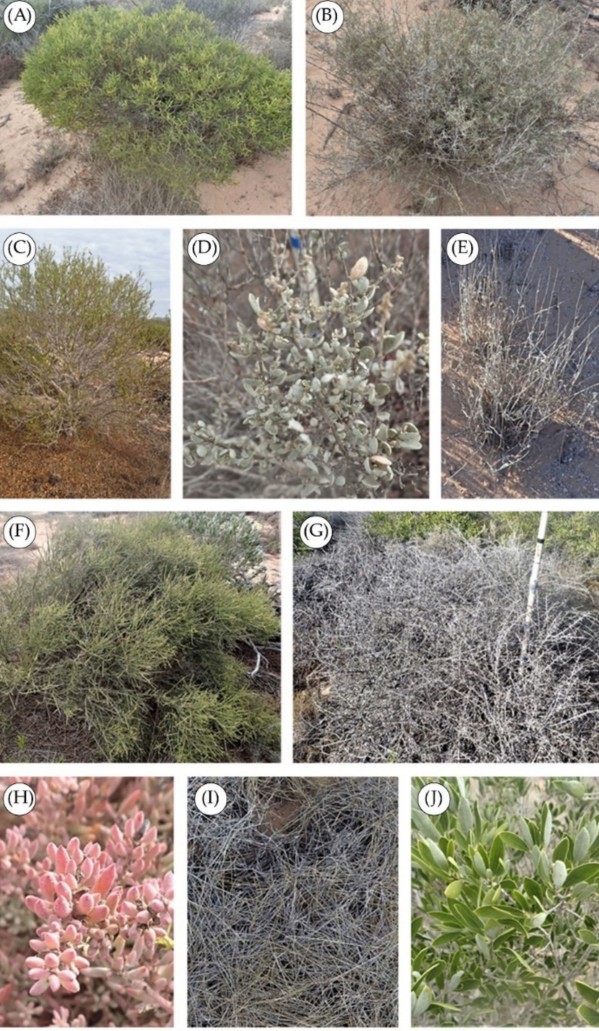

**Figure 2.** Common plants found on Dirk Hartog Island within the areas of interest. (**A**) *Acacia ligulata* (N = 25, 21), (**B**) *Acanthocarpus preissii* (N = 17, 0), (**C**) *Alyogyne cuneiformis* (N = 25, 0), (**D**) *Atriplex vesicaria* (N = 27, 20), (**E**) I *Cenchrus ciliaris* (N = 22, 0), (**F**) *Exocarpus aphyllus* (N = 19, 0), (**G**) *Scaevola spinescens* (N = 0, 21), (**H**) *Threlkeldia diffusa* (N = 13, 0), (**I**) *Triodia plurinervata* (N = 16, 20),

and (**J**) *Pittosporum phillyreoides* (N = 10, 0). All species except *S. spinescens* were found at study site 1. Study site 2 was considerably less diverse, comprising only *A. ligulata*, *A. vesicaria*, *T. plurinervata*, and *S. spinescens*. Numbers of species sampled are shown as (N = number at study site 1, number at study site 2).

### 2.3. Object-Based Variables

RPA imagery was segmented using eCognition version 9.5.1 (Figure 3 [41]). For each object, a suite of 21 variables were computed based on spectral reflectance (mean of the red, blue, and green bands), height (mean, median, minimum, maximum, and 90th percentile), texture (mean, correlation, contrast, homogeneity, and entropy), and shape parameters (roundness, compactness, length/width, and area). Additionally, the mean, median, minimum, maximum, and 90th percentile was calculated from the Green Leaf Algorithm [42]:

$$GLA = (2G - R - B)/(2G + R + B) \tag{1}$$

where G = green spectral reflectance, R = red spectral reflectance, and B = blue spectral reflectance. The GLA produces an image in the range of −1 to 1, where negative values are generally soil and values of 1 indicate green leaves and stems. Maximum, median, and mean GLA values were calculated for each segment. A canopy height model was produced for both study sites by subtracting an upper digital surface model from a digital terrain model.

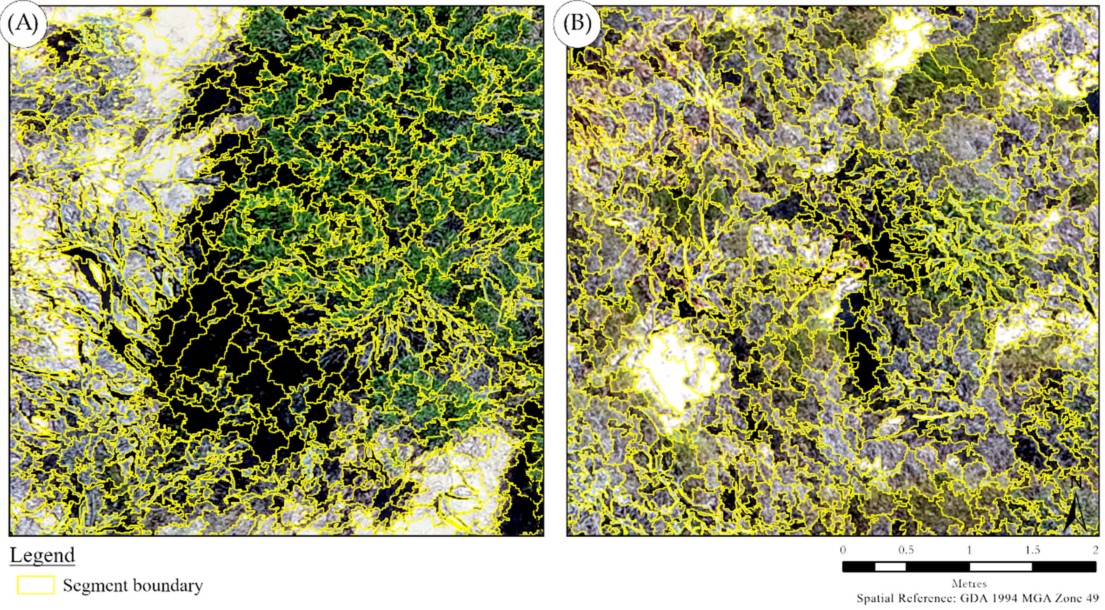

**Figure 3.** Detailed exemplars of multisegmentation for (**A**) study site 1 and (**B**) study site 2 quantified per the initial *c*. 5 mm imagery and a scale of 50.

Variable Importance in the Projection—VIP [43] scores were used to determine the subset of variables that could differentiate between the species found at each study site [6]. A VIP score ≤ 1 indicates that the variable is less likely to differentiate between species and may be excluded [44]. VIP scores using the preliminary segmented images were calculated for both sites.

### 2.4. Flight Height

To identify a suitable flight height for future surveys, we resampled our RPA imagery to mimic the capture resolution at higher altitudes. Both study sites were resampled using pixel sizes of 10, 15, 20, 25, 30, 35, 40, 45, and 50 mm. The resampled images and the initial *c*. 5 mm capture were segmented using 5, 10, 20, 50, 100, 200, 300, and 400 scale levels. The

accuracy of each pixel and segment scale combination was quantified per study site using the error estimate technique in lieu of a best-fit combination. Mean segment size (m$^2$) was plotted against Kappa to identify the optimal capture approach.

Spatial resolution is measured using Ground Sample Distance (GSD), which is the ground area from which reflectance is measured for an individual pixel. GSD here assumes pixel units are regular quadrilaterals and was measured based on the following equation:

$$GSD = \frac{FlightHeight \times SensorWidth \times 1000}{FocalLength * ImageWidth} \tag{2}$$

where GSD is the distance between the centre point of two consecutive pixels (mm/pixel); flight height is the distance (in metres) between the terrain and RPA lens; and sensor width, focal length, and image width are optical properties that vary with camera make and model. For the DJI Phantom 4 Pro used here, sensor width = 12.83 mm, focal length = 8.6 mm, and image width = 5472 pixels.

### 2.5. Classification

The Random Forest (RF) algorithm was implemented in R version 3.6.1 [31] using the ranger [37] and caret [38] library packages. Model constants applied to each iteration were mtry = default values 4 and 6, minimum node size = 1, and number of folds = 5. The mtry value determines the number of variables or predictors sampled at each tree split. Mtry values of 4 and 6 were trialled with the RF model operating on the best performing option. Minimal node size was set to 1 to ensure tree growth was not restricted, but this is the most computationally intensive choice. The number of folds parameter randomly portions 20% of the data per each iteration for an out-of-bag error estimate using a process known as bootstrapping. Out-of-bag bootstrapping 'drops' a percentage of testing data for iterations $n_1$, $n_2$ ... $n_{max}$ to determine accuracy [45].

An out-of-bag error was estimated from 100 iterations to quantify overall accuracy (OA) and Cohen's Kappa coefficient values. Kappa evaluates the agreement between random and observed class values [46]. Out-of-bag error estimation was complemented with an independent accuracy assessment using a stratified random testing subset of 30% of the sampled plant species. The independent cross-validation accuracy separated all segments within the plant canopy into testing and training subsets. Training and testing data partitioning was based on objects comprising the entire plant canopy, whereas the out-of-bag training used individual segments of the plant canopy.

The scale and pixel parameter combination that resulted in the highest accuracy value was used to calculate the final segmented outputs across both study sites, per the VIP subset variables. Error matrices, errors of omission, errors of commission, OA, and Kappa were computed for each site and calculated using the independent accuracy assessment previously described.

## 3. Results

### 3.1. Object-Based Variables

Both study sites returned eight variables that were likely to discriminate between all respective species, having a VIP score greater than 1 (Table 1). Whilst these differed in their importance, six of those variables were common to both sites. These were reflectance for each of the three RGB bands and three height variables (median, maximum, and 90% percentile). Study site 1 also identified the maximum GLA and 90th percentile GLA, but no texture or shape variables. The minimum height variable was also identified as discriminable in study site 2, along with one texture variable (Homogeneity). No other texture variables or shape variables were useful.

**Table 1.** Variables derived per object and Variable Importance in the Projection (VIP) scores quantified for study sites 1 and 2. The values highlighted in grey show the variables that were significantly differentiated (above VIP threshold for rejection).

| Variable Group | Variable | Study Site 1 VIP Score | Study Site 2 VIP Score |
|---|---|---|---|
| Reflectance | Mean red band | 1.76 | 1.17 |
| | Mean green band | 1.58 | 1.10 |
| | Mean blue band | 1.79 | 1.24 |
| Spectral index–Green Leaf Algorithm (GLA) | Mean GLA | 0.25 | 0.80 |
| | Median GLA | 0.06 | 0.56 |
| | Maximum GLA | 1.58 | 0.98 |
| | GLA 90th percentile value | 1.53 | 0.97 |
| Height–Canopy Height Model (CHM) | Mean CHM | 0.74 | 0.80 |
| | Median CHM | 1.18 | 1.43 |
| | Minimum CHM | 0.88 | 1.01 |
| | Maximum CHM | 1.35 | 1.68 |
| | CHM 90th percentile value | 1.30 | 1.61 |
| Texture | Mean | 0.11 | 0.79 |
| | Correlation | 0.65 | 0.68 |
| | Contrast | 0.35 | 0.76 |
| | Homogeneity | 0.52 | 1.24 |
| | Entropy | 0.51 | 0.74 |
| Shape | Roundness | 0.02 | 0.53 |
| | Compactness | 0.00 | 0.51 |
| | Length/width | 0.04 | 0.27 |
| | Area per pixel | 0.03 | 0.75 |

*3.2. Flight Height*

The maximum OA and Kappa coefficient obtained for the study site 1 out-of-bag error assessment were 0.58 and 0.48, respectively (Figure 4A,C). The independent error assessment returned a slightly lower OA and Kappa of 0.51 and 0.41, respectively (Figure 4B,D). Isolines plotted against segment scale over size heat maps show an accuracy hot spot following the diagonal intersection between segments computed with a scale of 20 and pixel size of 5 mm to a scale of 5 and pixel size of 45 mm. Australian RPA flights are restricted to 120 m above ground [47], which would result in an optimal pixel size of 33 mm per the GSD equation.

The diagonally accurate values pattern was also found for study site 2. This site returned higher maximum independent (OA = 0.67 and Kappa = 0.56) and out-of-bag (OA = 0.71 and Kappa = 0.62) accuracies when compared with study site 1.

According to Landis and Koch [48], Kappa values between 0.4 and 0.6 equate to a moderate reliability for study site 1 (out-of-bag Kappa = 0.48) and a 0.6–0.8 Kappa range shows that study site 2 has substantial reliability (out-of-bag Kappa = 0.62).

The optimal mean segment area per the out-of-bag error assessment was found to be 0.0116 m$^2$ (Kappa = 0.48) and 0.0112 m$^2$ (Kappa = 0.62) for study sites 1 and 2, respectively, with a scale of 5 and pixel size of 40 mm for both.

The mean segment area quantified for study site 1 per the independent assessment achieved optimum at 0.0361 m$^2$ (Kappa = 0.41) with a scale of 10 and pixel size of 30 mm. Mean segment area quantified for study site 2 per the independent assessment achieved optimum at 0.0162 m$^2$ (Kappa = 0.56) with a scale of 5 and pixel size of 45 mm.

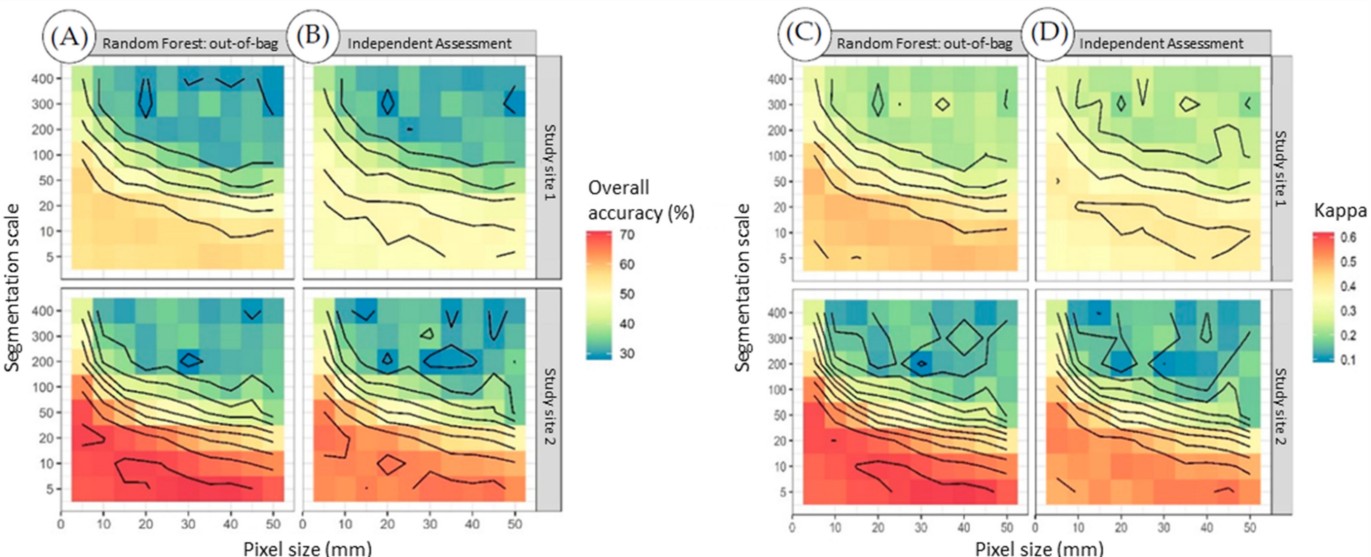

**Figure 4.** Segmentation scale over pixel size (mm) heat maps with isolines for each study site showing (**A**) out-of-bag overall accuracy, (**B**) independent overall accuracy per a 30% split of the data, (**C**) out-of-bag Kappa, and (**D**) independent Kappa per a 30% split of the data.

### 3.3. Classification

Errors of omission, also referred to as type II errors, represent false-negative samples that were omitted from their correct class. Errors of commission, or type I errors, refer to false positives where samples are incorrectly included in a class and thus the true null hypothesis is rejected. High errors of omission and commission across study site 1 were found for *A. preissii*, *A. vesicaria*, *T. diffusa*, *T. plurinervata*, and *P. phillyreoides* (Table 2A). *C. ciliaris* and *E. aphyllus* also showed high errors of commission (80.90% and 60.66%, respectively) for study site 1.

Study site 2 had low to moderate errors of omission for all species (Table 2B). A high error of commission was found for *A. vesicaria* (74.17%).

The final outputs for study sites 1 and 2 showed overall classification accuracy values of 53.03% and 70.24%, respectively (Figure 5A,B).

**Table 2.** Confusion matrix calculated with errors of commission (Comm) and omission (Om) shown for (**A**) study site 1 and (**B**) study site 2.

| | | *A. ligulata* | *A. vesicaria* | **Ground** | *T. plurinervata* | *S. spinescens* | *A. preissii* | *A. cuneiformis* | *C. ciliaris* | *E. aphyllus* | *T. diffusa* | *P. phillyreoides* | Comm Error (%) |
|---|---|---|---|---|---|---|---|---|---|---|---|---|---|
| (A) | *A. ligulata* | 268 | 31 | 54 | 3 | - | 23 | 123 | 2 | 89 | 9 | 4 | 55.8 |
| | *A. vesicaria* | 28 | 59 | 129 | 11 | - | 26 | 33 | 1 | 10 | 26 | 2 | 81.8 |
| | Ground | 12 | 22 | 838 | 7 | - | 16 | 18 | 2 | 14 | 11 | 1 | 10.9 |
| | *T. plurinervata* | 4 | 8 | 48 | 14 | - | 12 | 17 | 1 | 0 | 19 | 0 | 88.6 |
| | *S. spinescens* | - | - | - | - | - | - | - | - | - | - | - | - |
| | *A. preissii* | 17 | 30 | 60 | 2 | - | 62 | 35 | 0 | 1 | 10 | 3 | 71.8 |
| | *A. cuneiformis* | 55 | 12 | 53 | 2 | - | 10 | 412 | 2 | 27 | 9 | 1 | 29.2 |
| | *C. ciliaris* | 1 | 4 | 37 | 5 | - | 5 | 6 | 17 | 1 | 13 | 0 | 80.9 |
| | *E. aphyllus* | 162 | 6 | 1 | 1 | - | 9 | 53 | 1 | 155 | 4 | 2 | 60.7 |
| | *T. diffusa* | 2 | 14 | 58 | 9 | - | 12 | 30 | 2 | 3 | 11 | 0 | 90.5 |
| | *P. phillyreoides* | 10 | 4 | 0 | 0 | - | 6 | 18 | 0 | 0 | 4 | 1 | 97.7 |
| | Om error (%) | 52.1 | 68.9 | 34.4 | 74.1 | - | 65.8 | 44.7 | 37.0 | 48.3 | 90.5 | 92.9 | |
| | Overall accuracy (%) | 53.0 | | | | | | | | | | | |
| (B) | *A. ligulata* | 1274 | 26 | 25 | 80 | 248 | | | | | | | 22.9 |
| | *A. vesicaria* | 34 | 70 | 0 | 28 | 139 | | | | | | | 74.2 |
| | Ground | 4 | 0 | 212 | 7 | 0 | | | | | | | 3.6 |
| | *T. plurinervata* | 16 | 1 | 34 | 317 | 2 | | | | | | | 14.3 |
| | *S. spinescens* | 145 | 64 | 18 | 50 | 294 | | | | | | | 48.51 |
| | Om error (%) | 13.3 | 56.5 | 26.6 | 24.2 | 56.9 | | | | | | | |
| | Overall accuracy (%) | 70.2 | | | | | | | | | | | |

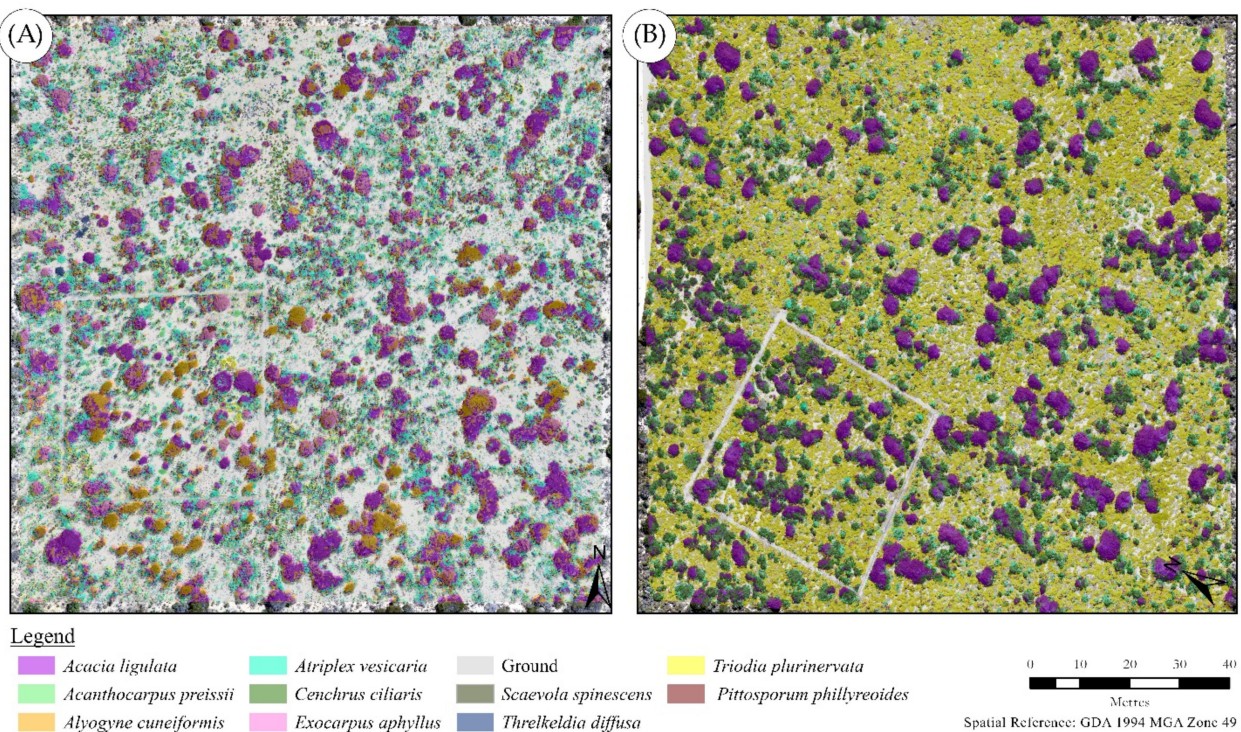

**Figure 5.** Study site 1 (**A**) and study site 2 (**B**) maps showing predicted segments quantified per a scale of 5, pixel size of 45 mm, and each individual VIP variable subset.

## 4. Discussion

Achieving an affordable, replicable, and accurate method for plot-scale monitoring is crucial to ensure that conservation actions are successful (Buters et al., 2019). There have been successful classification studies identifying larger growth forms using canopy delineation from RPA imagery [49]. For example, Baena, Moat, Whaley, and Boyd [7] showed that RPAs could accurately identify 95.3% of dry forest tree species in San Francisco de Asis, Northern Peru, each with the segment bounded to individual canopies. However, this study showed similar morphologies with no canopy mixing between the tree species (*n* = 3) being measured and thus entire canopies could be differentiated. Furthermore, restoration efforts can result in successional vegetation changes and require mapping of both pioneer and climax plant species with varied growth forms [50]. To overcome the challenge of identifying rangeland plants with coexisting plant species and growth forms (grasses to large shrubs), this study modelled the segment closer to the leaf size than the canopy size. The optimal segment size was found to be approximately 0.01 m$^2$, which achieved an overall detection rate greater than 70% when species diversity was low (*n* = 4).

Significantly important variables identified were derived from visible bands (*n* = 6) and height variables (*n* = 3). Spectral reflectance and height are commonly used in imagery classification, and our results support traditional remote sensing practices (e.g., [10,51,52]). Study site 1 also recognised two variables based on the GLA as important for differentiating plants, which follows the logic that spectral indices can be used to separate different plant species [53]. However, there were no texture or shape variables identified as important for study site 1, which are argued to be an advantage of OBIA [54,55]. Furthermore, study site 2 returned a sole texture variable and no shape variables. Studies examining land use and land cover show texture and shape variables can be valuable at a regional scale [56–58]. Our results suggest that in plant-species-level classifications, texture and shape may provide less information on the degree of separation between classes.

Another advantage of OBIA is the ability to mitigate the 'salt-and-pepper' phenomenon which can occur in traditional per-pixel methods [59]. This is when individual

pixels cannot be classified due to the sensor's instantaneous field of view being smaller than the size of the object being recorded or the surface being highly heterogenous [60]. Whilst RF did assign a class to each segment, close inspection of the canopies showed that there was possible noise present within the plant boundaries of multisegmented individuals. The intraclass heterogeneity and/or interclass similarities may have resulted in inaccuracies when determining a species-level classification. There is the potential to predict the correct class of erroneous segments postclassification based on peripheral segments to improve the classification accuracy [61]. As it stands, there are limited studies applying a multisegmentation approach for species-level classification using an affordable RPA, and thus the refinement of the technique has yet to be achieved.

Comparisons between pixel size, segment scale, and Kappa values showed that a finer spatial resolution does not always result in a higher accuracy. Studies examining the accuracy achievable in relation to resolution have shown varying results (e.g., [62,63]. Our results show that a pixel size of 45 mm appears just as capable as 5 mm. Increasing the capture altitude can expedite survey time and capture a greater area of imagery [64]. The maximum regulatory flying height of 120 m [47] can achieve an optimal pixel size of 33 mm on Dirk Hartog Island. Therefore, with the increase in altitude corresponding to an optimal accuracy estimate, conservation practitioners can achieve a greater image coverage and thus an increase in survey efficiency. Aerial surveys for repeat environmental monitoring should consider whether the flight plan could be enhanced by increasing the flying altitude whilst maintaining an optimal accuracy.

In this study, we used variance importance in the projection (VIP) for dimensionality reduction. A measure known as the Bhattacharyya distance is another possibility for seeking the best combination of variables and has been shown to be a robust measure in rangelands [65]. Metrics of under- and over- segmentation could be utilised to complement the use of Kappa used here for exploring the optimal segmentation scale. Under-segmentation occurs when the segment exceeds the unit of reference and over-segmentation results from the unit of reference being excessively divided [66].

Testing for errors of commission and omission indicated that there was a greater level of plant species confusion for study site 1 when compared with study site 2. Increased species diversity may result in difficulties for machine-learned classification of standard RPA imagery. Literature applying RF to successfully classify vegetation has either grouped species into plant communities (e.g., [67]) or analysed fewer species in the landscape (e.g., [68]. Species such as *A. preissii*, *A. vesicaria*, *T. diffusa*, and *T. plurinervata* were omitted from their true class whilst some, e.g., *C. ciliaris* and *E. aphyllus*, were inaccurately included in false classes for study site 1. *A. preissii* and *T. diffusa* are both low-lying perennial herbs with similar branching [69,70]. Another example of species which share similar growth forms are the grasses *T. plurinervata* and *C. ciliaris*. Grouping herbs and grasses into a single class can still provide information on plant responses after species translocation and may improve the accuracy of environmental monitoring using RPA imagery [71,72]. If feasible, it is suggested to group species with similar characteristics (e.g., growth form) to reduce classifier confusion [73]. Another reason classes can be confused in species-diverse sites is that finer variation in spectral reflectance, such as the greyish hues present in *A. vesicaria* leaves [74] or verdant chlorophyll pigmentation of *E. aphyllus* [75], may not be measured using course multispectral imagery. Increasing the spectral resolution, through splicing bands or increasing the number of bands captured, could allow for the differentiation of species with diverse plant communities. For example, near-infrared is a useful band for measuring vegetation [4]. Lastly, species such as *C. ciliaris* and *S. spinescens* can show different colouration after rain. Timing the capture per the season may provide more variation between species, thus improving classification. This highlights the need to ensure consistent timing if undertaking repeat monitoring.

This paper has shown that affordable RPA capture is viable for repeat monitoring of plant species on Dirk Hartog Island where species diversity is low. The translocation of native animals to Dirk Hartog Island was planned with the aim to quantify their restoration

of biodiversity and ecosystem services [25]. Measuring vegetation in areas excluded from reintroduction can be compared to areas where fauna is permitted for the purpose of long-term ecosystem monitoring. Momentum for protecting native fauna in Australia is currently shifting towards the aim of ecosystem restoration, with 74% of post-2018 digging mammal translocations stating this as a goal [29]. Bioturbation by a digging mammal can reduce soil hydrophobicity [28]; increase soil nutrients [76]; improve soil density [77]; and influence seed germination [28]. These qualities can restore plant species' heterogeneity and ecosystem function, thus resulting in landscape changes over time [78,79]. Optimisation of the Dirk Hartog Island RPA capture using OBIA and RF will provide conservation practitioners a best-practice guideline to assist with measuring the efficacy of restoration efforts.

## 5. Conclusions

Very high resolution RPA imagery analysed using the DJI Phantom 4 Pro and an OBIA is viable for repeat vegetation monitoring in rangeland environments, including at the species level if species diversity is sufficiently low. Common imagery variables between the study sites found spectral reflectance bands and canopy height model variables to be the most useful for detecting plant species, whilst measures of shape and texture were not. Results show that the 45 mm image pixel size can be as capable as a 5 mm pixel in classifying vegetation. This suggests that a higher spatial resolution does not necessarily result in a greater classification result. Optimal flight height for monitoring exclusion plot sites was considered to be the regulatory maximum of 120 m above ground, corresponding to a spatial resolution of 33 mm. This would allow for a greater survey area to be captured for future monitoring surveys whilst maintaining optimal accuracy. Future environmental monitoring using RPA imagery to detect plant species may improve the accuracy obtainable by grouping like species or applying a postclassification probability filter. Increased spectral bands, such as near-infrared, would also likely increase classification accuracy. The methods and data presented here provide an important guideline to understand plant responses from translocated animals on Dirk Hartog Island.

**Author Contributions:** Conceptualisation, R.v.D.; methodology, R.v.D., L.W. and T.P.R.; software, R.v.D. and L.W.; validation, L.W., R.v.D. and T.P.R.; formal analysis, L.W., R.v.D. and T.P.R.; investigation, L.W., R.v.D. and T.P.R.; resources, L.W., R.v.D. and S.C.; data curation, R.v.D.; writing—original draft preparation, L.W.; writing—review and editing, L.W., R.v.D., S.C. and T.P.R.; visualisation, L.W. and T.P.R.; supervision, T.P.R.; project administration, L.W. and R.v.D.; funding acquisition, R.v.D. and S.C. All authors have read and agreed to the published version of the manuscript.

**Funding:** This research was funded by the Gorgon Barrow Island Net Conservation Benefits Fund, administered by the Department of Biodiversity, Conservation, and Attractions (DBCA) and approved by the Minister for Environment.

**Data Availability Statement:** For data accessibility please contact the corresponding author.

**Acknowledgments:** This study acknowledges the DBCA, Government of Western Australia. In particular, the authors would like to thank Katherine Zdunic for assisting with in situ field surveys and Paul Rampant for his support with RPA imagery processing as well as Adam Cross from Curtin University and Vanessa Westcott from Bush Heritage Australia for their help identifying plant samples. Finally, we are grateful for Daniel McIntyre's helpful review of our manuscript prior to submission.

**Conflicts of Interest:** The authors declare no conflict of interest. The funders had no role in the design of the study; in the collection, analyses, or interpretation of data; in the writing of the manuscript; or in the decision to publish the results.

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
