# Peer review of "Mapping Restoration Activities on Dirk Hartog Island Using Remotely Piloted Aircraft Imagery"

_remotesensing, doi:10.3390/rs14061402_

Round 1

Reviewer 1 Report

The authors have done an interesting research research using low cost drones. All my comments in the first review have ben addressed.

Author Response

Thank you. We are delighted to hear this.

Reviewer 2 Report

This manuscript demonstrates the use of object based image analysis to classify UAS images taken at two different elevations. The manuscript is a nice application; however, it lacks innovation since most of the results/conclusions are well-known for the research community. I suggest that the authors do a comprehensive literature review on using OBIA for natural area classification using high resolution imagery. Below are specific comments related to the manuscript:

  1. The introduction section should introduce the rich literature in this subject and identify the specific merit of this study.
  2. All the variables in equation 2 must have units
  3. The field training data were collected using tablet with know specified positional accuracy. This is important given the mm level pixel size of the produced images.
  4. What are the variables extracted and used to select the best variables using the VIP technique?  Table 1 should be introduced in the Material and Methods section with a description of the used variables. It is interesting that there is no mention of how the plant height variables were obtained.
  5. The size of the training data (and hence the validation data since it is a percentage of the training data) needs to be specified for each class. 
  6. Please include all the parameters used in the RF classification algorithm. Typically the maximum number of trees are specified. Also, please mention the used software in the same place the library package is mentioned.

Author Response

  1. The introduction section should introduce the rich literature in this subject and identify the specific merit of this study.

Thank you. We have updated our review on OBIA in the introduction. Please see Section 1.2. We have tried to be succinct and to the point that OBIA is a superior technique to pixel-based methods.

  1. All the variables in equation 2 must have units

Thank you. We have ensured all units are listed in the parameters.

Specifically - Where GSD is the distance between the centre point of two consecutive pixels (cm/pixel), flight height is the distance (in metres) between the terrain and RPA lens; Sensor width, focal length and image width are optical properties that vary with camera make and model. For the DJI Phantom 4 Pro used here, sensor width = 12.83 mm, focal length = 8.6 mm, and image width = 5,472 pixels.

  1. The field training data were collected using tablet with know specified positional accuracy. This is important given the mm level pixel size of the produced images.

Absolutely, yes. If we were using only the GPS of the tablet, that would be an issue. We have tried to clear this confusion up. The tablet was only used as an assistive tool in the field. We have reworded to this:

“A Samsung Android tablet displaying the georectified RPA images as an assistive tool for mobile field collection. Plants that could be unambiguously matched in the field to the image were delineated and each individual was photographed and given an incremental photo number.”

  1. What are the variables extracted and used to select the best variables using the VIP technique?  Table 1 should be introduced in the Material and Methods section with a description of the used variables. It is interesting that there is no mention of how the plant height variables were obtained.

The variables used are described on lines 211-216 and highlighted in grey in Table 1. How Table 1 was derived is supplied on lines 153-156 (methods section).

The plant height variables are essentially derived as a canopy height model (or normalised DSM) produced by subtracting the DSM from the DTM. Please see lines 149-152.

  1. The size of the training data (and hence the validation data since it is a percentage of the training data) needs to be specified for each class. 

Thank you – we have reworked the caption of Figure 2 to include the number of species sampled from each group. We assume you mean that rather than the number of segments, which can be found in the confusion matrix.

  1. Please include all the parameters used in the RF classification algorithm. Typically the maximum number of trees are specified. Also, please mention the used software in the same place the library package is mentioned.

Thank you. We have noted that the RF classification was conducted in R 3..6.1 using the caret and ranger packages. We have supplied all parameters we have used to allow. We have not restricted the ntree parameter to any number. The reasoning for setting minimal node size to 1 was to avoid restricting tree growth at all. We have made minor modifications to the text around this point.

Reviewer 3 Report

  1. Literature review for the object based image analysis is too weak.
  2. Resolution of all figures need to be improved.
  3. In general, the work carried out in this draft is interesting, the only issue that worries me is the novelty of this work.

Author Response

  1. Literature review for the object based image analysis is too weak.

Thank you. We have updated our review on OBIA in the introduction. As noted above, we have still tried to remain succinct here, as it is becoming very well known that OBIA is generally superior to pixel-based techniques.

  1. Resolution of all figures need to be improved.

Resolution has been improved to be >300 dpi and up to 1000 dpi. 

In general, the work carried out in this draft is interesting, the only issue that worries me is the novelty of this work.

Thank you. There is no doubt a plethora of work involving drones as they are still finding their niche between satellite and larger airborne platforms. The major outcome of this research shows that relatively cheap platforms are all that are needed for monocultures and low diversity environments, but as diversity increases improvements in sensor technology are warranted. This is an exciting find for our research as it allows us to monitor and compare plots included and excluded from reintroductions and translocation of ecosystem engineers.

Round 2

Reviewer 2 Report

This manuscript presents a typical remote sensing image classification application using object based analysis approach and machine learning techniques. Although the application is presented nicely, it lacks novelty. The analysis methods presented in this study could be appropriate for publication in a remote sensing journal 5-10 years ago. The content could be suitable for publishing in other journals interested in the outcome of the classification (e.g. ecological or forestry journals).

Author Response

Thank you for you thorough review. Whilst OBIA has been previously published, the novelty in this work is that we present a study which shows that a higher spatial resolution will not always result in a greater accuracy. Furthermore, we attempt to classify a diverse community of rangeland plants to a species level using multi-segmentation. This is an ambitious task and, to my knowledge, has not been scientifically examined.

Reviewer 3 Report

No more comments.

Author Response

Thank you for your review

This manuscript is a resubmission of an earlier submission. The following is a list of the peer review reports and author responses from that submission.

Round 1

Reviewer 1 Report

“” Plots had differing species rich-15 ness and diversity allowing for the comparison of discriminatory variables, classification potential, 16 and optimal flight height””

Seems a  too complex sentence please try an edit or cut in smaller sentences

 We found visible bands and height variables to have high discrimination 17 potential and measures of texture to be poor discriminators 

Idem, very compact sentence . Maybe split it in 2 or 3 small sentences be aware texture is useful in many cases, thus it is rare that here it is not useful.  

“”

 Remote sensing as an assistive dataset for testing conservation efficacy has been largely 28 untapped due to previous impediments such as pixel size.””

Pixel sizes of 10 cm is already known from Photogrammetry airborne platforms since long time. RS with satellites is here intended ???

“”

 relative to coexisting species is important for tracking the trajectories 37 of individual species””

do we discuss spectral profiles over time here ??

“is not feasible where plant species co-exist and growth forms intertwine. We propose that 46 multi-segmentation of the individuals within an image may overcome this challenge. 47..........................

Dirk Hartog Island, Western Australia pastoralists used to farm sheep between c. 48 1869 to 2009”

Please make a sub section here. The OBIA and the history of the island should be separated well

“”Aerial imagery (RGB) was captured using a DJI Phantom 4 Pro RPA on 16th Septem-98 ber, 2018 with a radiometric resolution of 8-bits.””

Old infrared imagery is available in all area’s flown by the RAF since WW II   no such data was taken into account ?? Scanned infrared imagery from 20 to 50 years ago are a real treasure for such monitoring. It also allows to demonstrate the difference with less disturbed climax situation of those species involved

Please also look into similar: https://pubs.usgs.gov/of/2017/1093/ofr20171093.pdf

“”Figure 2. Common plants found on Dirk Hartog Island within the areas of interest” Difference in texture seems evident in the pictures. Please cross check on the textural analysis. Maybe with other peer groups

“species. The independent cross-validation accuracy separated 179 all segments within a plant canopy into testing and training subsets”

It seems “over segmentation” of shrub species would result in a population of sub-objects for each individual shrub crown. Please show a typical situation of a non-pioneer, closer to climax or steady state situation ( if exist) maybe elaborate on the development stage . Is climax of this vegetation known and still existing? or all of these species are only found in deteriorated environments ??

What is “on average” the amount of segments for each individual crown  ? if over 25 segments, also standard deviation can be investigated as textural measurement.

Talking about “”Plants” might also include plants that do not have a woody architecture and remain close to the surface.

“”These were the simple mean 192 of the three RGB bands and three height variables (median, maximum, and 90% percen-193 tile).””

That is strange, normally the mean needs to be normalized over the total intensity. Sure no normalization was improving the mean ??  ( such as ratio to scene for example)

Figure 4 is very useful ! compliments.

Indeed Infrared is useful

But modern applications involve repeated data collection as well

A scenario more closer to present state of agricultural monitoring might be an option

In your region , among others, this is practiced by:

https://aspioneer.com/brooke-tapsall-an-innovator-in-the-drone-industry/

and

https://www.linkedin.com/company/seequent/?originalSubdomain=sv

Reviewer 2 Report

This manuscript, titled "Monitoring of restoration activities on Dirk Hartog Island using remotely piloted aircraft imagery" tests the ability of UAV (DJI Phantom 4 Pro) imagery to discriminate plant species in two plot sites on Dirk Hartog Island. Object based image analysis method and RF algorithms were used to classify plant species. It is an interesting article about using the low-cost UAV in vegetating mapping, and I believe it adds technical knowledge to remote sensing applications. Following specific comments that may help authors to improve this manuscript.

  1. In title. It’s strange the word monitoring appears here. I think what the authors do is just mapping.
  2. In 2.4. The authors did a resampling exercise to obtain multi-resolution images. Then, the comparison classification tests were performed using these data. However, there are still differences between these data and data collected directly by drones in the field. With the altitude of the UAV changes, not only does the spatial resolution of the UAV image change, but also the image coverage increases. I strongly suggest that the authors add this aspect, which should at least be mentioned in the discussion, that the efficiency of data collection increases as the altitude changes.
  3. In 2.5. As the authors report, the precision of the classification results in this paper is not very high. Although we generally agree that Random Forest is one of the best classifiers. However, I think a comparison of at least two classification methods should appear in this study.
  4. In 5. The conclusions are weak compared to the level of research in the paper. There is no information relating to the variables of the UAV images.

Reviewer 3 Report

This manuscript presents the use of high-resolution RGB images captured by a camera mounted on a DJI phantom 4 small unpiloted aircraft. The manuscript adopted an object-based analysis approach on a limited number of classes. I think this research lacks novelty and merits. Given the rich literature utilizing the same image types and analysis methods, I do not see an addition to the body of knowledge in this field. I wished to see more review/comparison with existing similar literature, some of which achieved much higher accuracies. The low classification accuracies reported in this study and some of the conclusions, such as the low importance of textural features, contradict a wider consensus of the importance of texture information, especially with the poor (3 bands) spectral information utilized in the research. In this type of situation the researchers are urged to thoroughly review their methods by for example, experimenting with more segmentation parameter choices. The lack of familiarity with the state-of-the-art literature in this area is clearly suggested by statements like 'As it stands there is an absence of studies applying a multi-segmentation approach and thus the refinement of the technique has yet to be achieved." in the manuscript.